The Author(s) *BMC Pregnancy and Childbirth* 2017, **17**(Suppl 2):342

**RESEARCH**

# A review of measures of women's empowerment and related gender constructs in family planning and maternal health program evaluations in low- and middle-income countries

Mahua Mandal[1*], Arundati Muralidharan[2] and Sara Pappa[3]

## Abstract

**Background:** Evidence suggests that gender-integrated interventions, which actively seek to identify and integrate activities that address the role of gender norms and dynamics, improve family planning (FP) and maternal health (MH). To understand the link between the gender components of interventions and FP and MH outcomes, it is critical to examine the gender measures used in evaluations.

**Methods:** We conducted a systematic review of evaluations of gender-integrated FP and MH interventions in low- and middle-income countries. We examine characteristics of the interventions and their evaluations, and summarize women's empowerment and related gender measures.

**Results:** Out of 16 evaluation articles, five reported the theoretical or conceptual model that guided the intervention. Twelve described how gender was quantitatively measured and identified 13 women's empowerment and related gender constructs. Gender scales or indexes were used in five evaluations, three of which noted that their scales had been validated. Less than one third of articles reported examining the effect of gender on FP or MH.

**Conclusions:** Evaluations of gender-integrated FP and MH interventions do not consistently describe how gender influences FP and MH outcomes or include validated gender measures within their studies. As a result, examining the pathways through which interventions empower women and the manner in which women's empowerment leads to changes in FP and MH outcomes remains a challenge. Valid measures of commonly reported women's empowerment and gender constructs, such as gender-equitable attitudes and women's decision-making power, must be adapted and used within evaluations to examine how empowerment and improvements in gender-related factors can produce positive FP and MH outcomes.

**Keywords:** Gender measures, Women's empowerment, Gender-integrated health interventions, Family planning, Maternal health, Low- and middle-income countries, Evaluations

* Correspondence: mmandal@email.unc.edu
[1]MEASURE Evaluation, Carolina Population Center, University of North Carolina at Chapel Hill, Chapel Hill, NC, USA
Full list of author information is available at the end of the article

## Background

Gender-integrated health interventions actively seek to identify and integrate activities that address the role of gender norms and dynamics. They take into account the potential effect of gender on proposed program objectives and the impact of program results on gender relations. Gender-integrated health interventions include those that are gender transformative and gender accommodating. Gender transformative approaches actively strive to challenge and change gender inequalities while promoting health. These approaches encourage critical awareness of gender roles and norms, challenge the distribution of resources and allocation of responsibilities between men and women, address power relationships between men and women, and promote the position of women. For example, a national policy may require women to be accompanied by their husbands to family planning clinics in order to get contraception. A gender transformative intervention would work to change this policy so that women can access contraception without their husbands' permission or presence. Alternatively, gender accommodating interventions work around inequitable gender norms, roles, and relationships or adjust for these inequalities. While these approaches do not actively seek to change norms and inequalities, they strive to limit the harmful impact of interventions on gender relations and the harmful impact of gender norms and inequalities on health outcomes [1, 2]. Using the same example above, a gender accommodating intervention would increase knowledge of the existing policy among couples in a community and encourage husbands to accompany their wives to clinics so that women can access contraception.

An increasing number of gender-integrated health interventions have been implemented over the past decade to counter deeply rooted gender inequalities in order to improve family planning (FP) and maternal health (MH) outcomes [2, 3]. Concurrently, donors and national governments have focused on evaluating these interventions to determine their level of success and potential for scale-up. Evidence suggests that gender-integrated interventions improve FP and MH outcomes [3]. The Sustainable Development Goals (SGDs), building upon the Millennium Development Goals, provide impetus for continued action on gender equality by tackling inequalities and empowering women and girls. Goal 5 of the SDGs explicitly links gender equality to health and well-being through the key target (target 5.6) to ensure universal access to sexual and reproductive health and rights [4]. Given the greater programmatic attention and global political will to improve gender equality for health, especially women's health, it is necessary to examine how gender is measured, particularly in terms of women's empowerment, when evaluating gender-integrated health interventions. This in turn will help development researchers and practitioners identify and act on the pathways by which empowerment contributes to FP and MH outcomes.

Conceptualizations of women's empowerment often highlight one or more of a series of interconnected concepts of choice, options, control, and power — concepts that allude to "women's ability to make decisions and affect outcomes of importance to themselves and their families" [1]. Keller and Mbewe [5] offer a comprehensive definition of women's empowerment encompassing these concepts: "a process whereby women become able to organize themselves to increase their own self-reliance, to assert their independent right to make choices and to control resources which will assist in challenging and eliminating their own subordination". Other academics have focused on one central concept; for example, Gita Sen's [6] definition centers on altering the balance of power. In the era following the 1994 International Conference on Population and Development in Cairo, which articulated a people-centered approach to development, Kabeer's [7] description of empowerment as "the expansion in people's ability to make strategic life choices in a context where this ability was previously denied to them" has been widely accepted and used. Two salient characteristics of empowerment emerge as common across these definitions: that of empowerment as a process, and that of autonomy or choice.

There are several challenges to measuring women's empowerment. As a multidimensional latent construct, women can be empowered (or disempowered) within five broad dimensions: psychological, social (including familial), economic, legal, and political [8]. Recent work on measures has also attempted to capture empowerment within sexual relationships [9] and around contraceptive use, pregnancy, and childbearing [10], indicating the need to conceptualize a sixth dimension of health empowerment [8]. Recognizing various dimensions of empowerment is important, because empowerment in one dimension does not necessarily indicate empowerment in another [8]. Indicators of women's empowerment can also be organized according to the level or sphere of operation, at individual, household, and aggregate or population levels [4]. For instance, the individual level includes self-esteem and self-efficacy; the household level includes household decision-making and mobility; and at an aggregate level, Malhotra and colleagues identify indicators related to the labor market, education, and political and legal status. As with the six dimensions above, the process of empowerment may operate in some levels and not in others [1].

Women's empowerment is also contextually specific, in both place and time [8]. For example, restrictions on women's mobility, which is a measure of women's

empowerment in Demographic and Health Surveys, is a relevant construct in South Asia but not in sub-Saharan Africa [11].

Finally, there is limited understanding of whether and how empowerment and related gender constructs are measured within the context of gender-integrated FP and MH interventions. While published reviews have explored the relationships between women's empowerment and fertility [10], gender-based power and reproductive health outcomes [12], and women's empowerment and maternal and child health [8], we found no reviews of health program evaluations that have examined the measurement of women's empowerment and related gender constructs and their relationship to FP and MH outcomes. This review builds on previous work by examining in depth women's empowerment and related gender measures within outcome and impact evaluations of gender-integrated FP and MH interventions.

## Methods

This study draws from a broader, large-scale systematic review examining the influence of gender-integrated interventions on multiple health outcomes in low- and middle-income countries (LMICs) [3]. Using key search terms described elsewhere [3], we searched peer-reviewed and gray literature in electronic databases (e.g., PubMed, Population Information Online (POPLINE), Scopus) and sourced bibliographies (e.g., Bill & Melinda Gates Foundation), organizational websites, and relevant conference websites. The search yielded 2450 documents; 709 were excluded based on a title relevancy check.

To determine final relevancy for the remaining 1741 documents, we used the following inclusion criteria for review: (1) the intervention was gender-integrated, per the Interagency Gender Working Group's Gender Equality Continuum tool [13]; (2) the document was written in English; (3) the intervention was implemented in a LMIC, per the World Bank's definition; (4) methods and results of outcome or impact evaluations were discussed; (5) the intervention took place between January 1, 2008 to June 30, 2013; and (6) the evaluation measured FP or MH outcomes. For this review, we further excluded the gray literature, studies that included qualitative methods only, and evaluations that measured HIV or gender-based violence without also measuring FP or MH outcomes.

Original data abstraction methods are described elsewhere [3]. For this analysis, additional data were abstracted on the following: whether the article included in its theory of change or conceptual framework the role of gender in FP or MH outcomes; whether the evaluation measured a dimension of women's empowerment or other gender construct; the women's empowerment dimensions and related gender constructs measured; the level of operationalization of the construct (individual, couple, service delivery, community); the structure of the gender measures (single items, multi-item scale, multi-item index); whether the measures were validated in the current study or previous studies; and whether the gender measure mediated the relationship between the intervention and main FP or MH outcomes. We first describe the characteristics of the interventions and their evaluations. We then summarize the women's empowerment and related gender measures. Finally, we discuss measurement gaps in current evaluations of gender-integrated FP and MH interventions.

## Results

Of the 196 documents from the broader systematic review, 180 documents were excluded because they did not fit the inclusion criteria for this review (Fig. 1). The resulting 16 articles describe evaluations of six transformative interventions [14–19] and ten accommodating interventions [20–29] (Table 1). The evaluations were

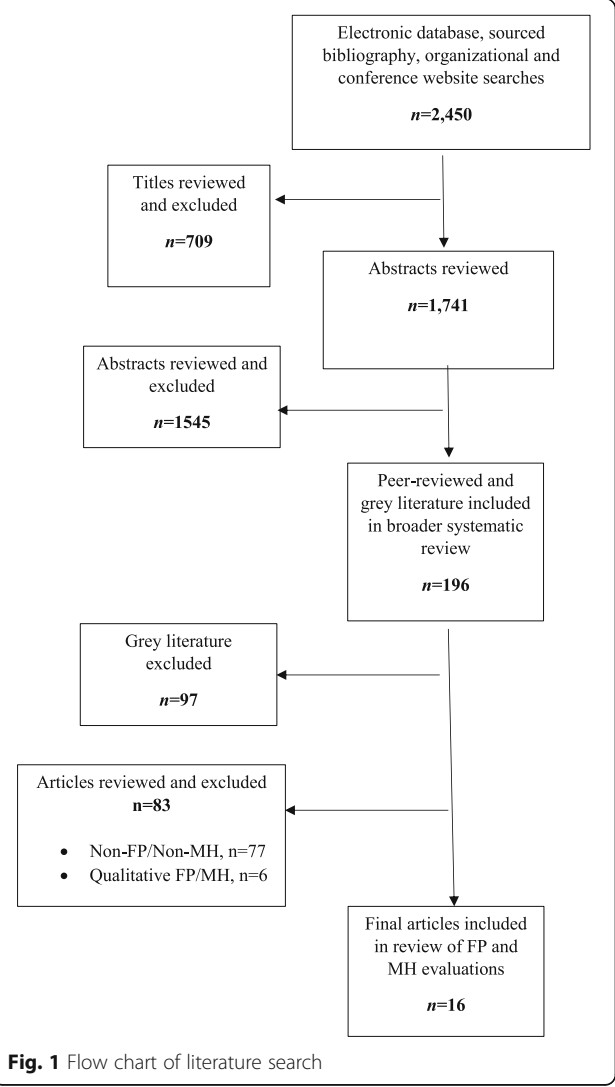

**Fig. 1** Flow chart of literature search

**Table 1** Women's empowerment constructs by level of operation and dimension of empowerment

| | Women's empowerment dimension | | | |
|---|---|---|---|---|
| | Psychological | Social | Economic | Health |
| Level of operation | | | | |
| Individual-level constructs | | | | |
| Confidence/self-esteem | X | | | |
| Gender-equitable attitudes (of woman or partner) regarding sexual and reproductive health (SRH) | | | | X |
| Gender-equitable attitudes (of woman or partner) regarding domestic matters | | X | | |
| Couple-level constructs | | | | |
| Communication around SRH matters | | | | X |
| SRH decision-making power | | | | X |
| SRH control | | | | X |
| Decision-making power regarding social life | | X | | |
| Support in pregnancy and health-seeking behaviors | | | | X |
| Support in childcare and housework | | X | | |
| Household-level construct | | | | |
| Domestic and financial decision-making power | | X | | |
| Service-delivery level constructs | | | | |
| Provider-client interaction | | | | X |
| Attitudes around women's health | | | | X |
| Community and societal-level constructs | | | | |
| Access to safe spaces | | X | | |
| Social networks | | X | | |
| Economic capabilities and assets | | | X | |
| Educational opportunities and participation | | | X | |

conducted mostly by international non-government organizations and span 15 LMICs (Bangladesh, India, Nepal, Pakistan, China, Mexico, El Salvador, Iran, Ethiopia, Tanzania, Malawi, Nigeria, Senegal, Tanzania, and Turkey) (see Fig. 2). Most interventions were implemented in rural areas; four were in urban areas [16, 22, 26, 29] and two in both rural and urban areas [19, 21]. Interventions often focused on women and men, as individuals, to increase knowledge, change attitudes, and improve health-related behaviors. Five interventions focused on individuals [16, 19, 21, 22, 26], and one intervened with couples only [29]. Several interventions engaged influential members of the household [20] and service providers [27, 28] to improve the supportive environment for women and men. Almost half the interventions ($n = 7$) intervened on multiple levels [14, 15, 17, 18, 23–25, 30]. The majority of interventions aimed to improve FP and MH outcomes of married adults. Only two interventions focused

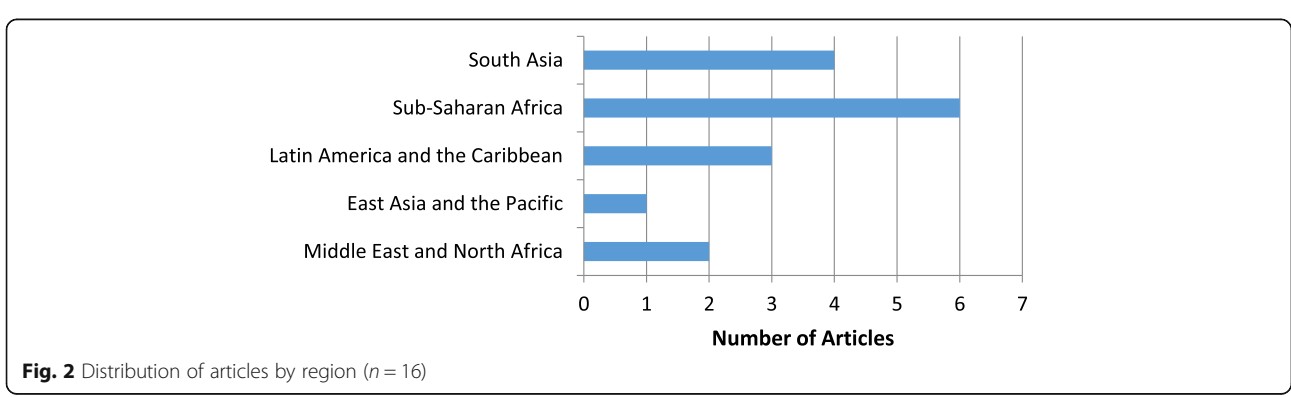

**Fig. 2** Distribution of articles by region ($n = 16$)

on adolescents: one intervened with married couples to delay pregnancy and childbearing [14]; another attempted to empower unmarried girls to delay marriage [15].

Few articles (two transformative [18, 19] and three accommodating [20, 23, 29]) reported the theoretical or conceptual model that guided their interventions. Both accommodating interventions were guided by two or more published theories. Theories and conceptual models described include the framework for enabling agentic empowerment; information-motivation-behavioral skills; health-seeking model; tri-sectoral health system; household production of health; theory of reasoned action; health belief model; and three delays model. No article clearly articulated a customized theory of change describing how various components of the intervention worked together in a causal pathway to achieve the intended outcome.

Only one article described its intervention as primarily focused on increasing women's autonomy [21]. While most discussed gender-related constraints and power imbalances that affect access to and use of health services, and described their interventions as either attempting to transform or accommodate prevailing gender norms, they did not explicitly name their interventions as women's empowerment interventions. Despite this, 12 interventions described (either a priori or in the discussion) interventions focused on improving FP and MH through strategies that empower women directly (e.g., education, livelihoods training, social support) [14, 15, 21] or indirectly (e.g., training service providers on respectful interaction with female clients, promoting gender-equitable attitudes, and emphasizing gender equity in decision-making among men) [16, 17, 27]. Three of the 12 were male involvement interventions that included supporting or empowering women through greater male partner engagement in domestic matters such as childcare and household chores [26] or in health matters [25, 26, 29]. Because we included evaluations of all gender-integrated FP and MH interventions (i.e., they were not limited to women's empowerment and health), our review consisted of two additional male involvement interventions that targeted males as the main beneficiaries [18, 24]. These interventions did not describe strategies used to empower women or promote gender equality.

All evaluations collected primary data, and their study designs were considered of high or moderate rigor, per criteria used in the broader systematic review [3]. The majority were randomized control trials or quasi-experimental studies. One study sampled separate cross sections of individuals at baseline and endline in the intervention area only [17]; another compared baseline and endline measures of the same group of individuals in the intervention area only [24]. All but one evaluation [25] collected data at two or more time points.

Half the evaluations measured FP outcomes [14–18, 21, 22, 27], two measured MH outcomes [20, 24], and six measured both FP and MH outcomes [19, 23, 25, 26, 28, 29]. The majority of evaluations of transformative interventions (*n* = 5) examined FP outcomes [14–18]; one examined both FP and MH outcomes [19]. Half of the evaluations of accommodating interventions (*n* = 5) measured both FP and MH outcomes [23, 25, 26, 28, 29], one measured FP outcomes [27], and two measured MH outcomes [23, 25].

Despite describing the prevailing gender norms and dynamics that affect maternal health, two evaluations that used quantitative methods only did not report measurement of women's empowerment or gender indicators [14, 29]. In two evaluations that used mixed methods, gender constructs emerged from the qualitative methods; gender indicators were not quantitatively measured [20, 24].

Our review identified 13 constructs within four dimensions of women's empowerment: psychological, social, economic, and health; with most constructs falling within social and health empowerment. The majority of constructs operate at the individual and couple level. Only one household-level and two service-delivery-level constructs were identified. The remaining three constructs fall within the community and societal levels (Additional file 1). Individual-level measures of women's empowerment were used in evaluations of interventions focused on the individual level [19, 22, 26], service-delivery level [28], and at multiple levels [15, 17, 18, 25]. A couple-level empowerment measure was used in one evaluation of an intervention operating at multiple levels [23]; a service-delivery level measure was used in an evaluation of a service-delivery intervention [27]; and household measures were used in evaluation of two individual-focused interventions [16, 21]. Gender-equitable attitudes, and support in pregnancy and health-seeking behaviors were the only constructs that were measured by more than one article. Less than half (*n* = 6) of articles included a single empowerment or gender construct.

Additionally, male involvement interventions not explicitly designed to empower women measured the following indicators and constructs: men's knowledge of reproductive and maternal health; men's awareness of their wives' use of MH services; attitudes towards gender norms; couple communication around FP; frequency of male communication about FP with partners, extended family, or other men in the community; joint decision-making around FP and MH; and equitable gender norms.

Few studies measuring women's empowerment or related constructs used scales that were validated in that

study (e.g., Cronbach's alpha) (Interpersonal Power, Relational Response to Condom Use, Comfort with Sexual Communication, Safer Sex Self-Efficacy) [16, 26] or in previous studies (Gender-Equitable Male scale and Supportive Behavior Questionnaire scale) [18]. One study reported using 13 scales but did not describe them in depth or report whether they were validated [19]. Two studies used indexes: Salutation, Assessment, Help, and Reassurance index [27] and Autonomy index [21]. The remaining studies used individual items or a series of items that were not combined into a composite or sum score. Four evaluations using validated scales or indexes were of FP interventions [16, 18, 21, 27], and one was of a combined FP and MH intervention [26].

Less than one third (*n* = 5) of studies reported examining the effect of gender on FP or MH outcomes. Four studies examined whether gender mediated the relationship between program participation (the main independent variable) and FP/MH outcomes [16, 18, 21, 27]. One study used separate statistical models to examine the effect of the intervention on the gender variable, and the gender variable on the FP outcome [21]. In two evaluations the gender variables were the same as or embedded within the dependent variable (e.g., men's awareness of their wives' use of antenatal care (ANC), attitudes around husbands participation in FP decisions) [22, 25]; thus, they were unable to examine the effect of women's empowerment or related gender constructs on FP and MH outcomes.

## Discussion

Despite restricting articles to gender-integrated interventions evaluations, few studies clearly articulated a hypothesized theory of change or (set of) causal pathway(s) between the intervention components, women's empowerment and related gender constructs, and FP and MH outcomes. Theories of change assist program planners and implementers in conceptualizing and understanding how elements of a program are related to one another and to the final desired outcomes; and they assist evaluators in identifying and mapping the constructs and indicators that should be measured to pre-specified components of the intervention [31]. This is particularly important in evaluations of gender-integrated health interventions, since women's empowerment and related gender constructs are multidimensional, operate at multiple levels, and affect access to services, health behaviors, and health outcomes in complex ways.

While no study included a comprehensive set, in terms of level of operation, of women's empowerment measures, most studies captured, at least partially, its key characteristics: autonomy and empowerment as a process. Measures reflected women's autonomy (e.g., decision-making power, economic capabilities and assets) or determinants

of women's autonomy (e.g., confidence/self-esteem). All evaluations except one measured changes over time, or the process of empowerment.

Surprisingly, none of the five evaluations of interventions in South Asia included women's mobility as a measure of empowerment, even though it is a commonly used indicator within research studies in the region [10]. This may be because the FP and MH interventions in the review's three South Asian countries did not attempt to address the lack of mobility as a barrier to FP or MH. Similarly, these evaluations did not include household-level measures of empowerment, even though parents and parents-in-law have substantial influence over a couple's family planning and childbearing practices, and reaching mothers- and fathers-in-laws was a strategy used in one intervention in India [14]. Such an exclusion results in incomplete mapping of the intervention components and strategies to appropriate indicators.

Some male involvement interventions designed to improve health through empowering their female partners also did not properly map indicators to program strategies. For example, the evaluation of an FP program emphasizing gender equality in decision-making among males measured two gender-related indicators: whether men communicated with their wives about FP, and men's attitudes around the role of men and women in FP decision-making. The first indicator measures merely the existence of couple communication and does not provide information regarding the quality of communication. Instances of communication may be characterized by coercion and male dominance rather than gender equality in decision-making. The second indicator more closely measures the desired concept, though it measures attitudes, not actual decision-making behaviors. Instead, the degree of joint decision-making (e.g., if the decisions are made mostly by the man, by the woman, or jointly; or who has the final say) is a more accurate measure of equality in decision-making.

Few articles described using validated measures of women's empowerment and related gender constructs, particularly evaluations of maternal health interventions; and several articles did not measure any gender construct. This finding is in contrast to the body of published studies of HIV and gender-based violence (GBV) program evaluations, which often use the Gender-Equitable Male scale [32–34] and Sexual Relationship Power Scale [35] to measure determinants or dimensions of women's empowerment. The lack of using validated measures in FP and MH interventions may be a result of FP and MH studies not adequately incorporating the latest knowledge from the SRH science base into evaluations of interventions.

Finally, we offer several considerations for the measurement of women's empowerment and related gender

constructs for future evaluations of gender-integrated interventions in LMICs. First, in order to measure the most pertinent forms of empowerment in FP and MH interventions, i.e., those related to reproductive health, we recommend that evaluations identify existing reproductive empowerment and related scales that have been developed in more developed countries, such as the Reproductive Autonomy and Reproductive Coercion Scales [36, 37], and test and adapt them to various developing country contexts. Researchers should be cognizant of the applicability and validity of such scales across regions and cultures that vary in their conceptualization of women's empowerment.

Second, given the limited number of interventions focused on adolescents, our review identified few empowerment measures specific to this population. The meaning and process of empowerment, particularly reproductive empowerment, depend on where an individual is in his or her life course, whether he or she is married or in a partnership, and whether a female is pregnant or a mother. Additional empowerment measures should be developed specifically for adolescents who are married, unmarried, nulliparous, pregnant, and mothers to better understand the pathways between adolescents' empowerment and positive FP and MH outcomes.

Third, the majority of existing measures of women's empowerment are at the individual and couple levels. Given that women can be empowered or disempowered at the household, service-delivery, and community and society levels, there is a need to develop measures at these higher levels and to map the level of the measures used in evaluations to the level at which interventions operate. In particular, multiple levels of empowerment measures should be used in evaluations of interventions that are multilevel.

Fourth, we note that our review did not identify measures of policies or laws that encourage women's empowerment or protect women's rights. Given the recent body of work on measuring health and human rights [38–40], we recommend that evaluations of gender-integrated FP and MH interventions take advantage of the momentum to develop measures of policies and laws in order to examine their impact on FP and MH. For example, FP evaluations may consider developing indicators to measure the number, type, and scope of policies and laws that hinder women's rights to FP, including availability, accessibility, acceptability, and quality of contraceptives, and participation in decision-making [41].

Fifth, the FP and MH evaluation field should look to studies of male-focused gender transformative HIV and violence interventions [42, 43] to identify and adapt measures of male engagement and related gender constructs. Measures of normative change among males, including attitudes around gender roles and masculinity, may be particularly important in evaluations of FP and MH interventions that involve men with the explicit purpose of increasing gender equity and women's empowerment.

Next, to measure the added value of gender in a program, evaluations must examine the relationship between participating in the program and a gender construct; and the articulated gender construct and the desired health outcome. This can be done statistically by including gender variables as mediators in regressions models. Psychometric statistical methods, such as structural equation modeling, may be particularly well suited for evaluations of complex interventions that aim to examine the relationship between women's empowerment, other latent constructs, and FP and MH outcomes.

Finally, given the inter-relatedness of different dimensions of empowerment, evaluations may consider including multidimensional scales of empowerment (e.g., economic, social, and reproductive) within single studies. Including a broader construct with multiple women's empowerment dimensions within evaluations can help identify dimensions most malleable and predictive of positive FP and MH outcomes. Program planners and implementers can use the results to further support and improve relevant forms of empowerment in FP and MH interventions and policies.

## Conclusions

Evaluations of gender-integrated FP and MH interventions do not consistently describe how gender influences FP and MH outcomes or include validated gender measures within their studies. As a result, examining the pathways through which interventions empower women and the manner in which women's empowerment leads to changes in FP and MH outcomes remains a challenge. Valid measures of commonly reported women's empowerment and gender constructs, such as gender-equitable attitudes and decision-making power, must be adapted and used within evaluations to examine how empowerment and improvements in gender-related factors can produce positive FP and MH outcomes.

### Open peer review

Peer review reports for this article are available in Additional file 2.

## Additional files

**Additional file 1:** Evaluation papers included in systematic review. (DOCX 27 kb)

**Additional file 2:** Open peer review. (PDF 129 kb)

## Abbreviations
FP: Family planning; HIV: Human immunodeficiency virus; LMIC: Low- and middle-income countries; MH: Maternal and child health; SDG: Sustainable development goal; USAID: United States Agency for International Development

## Funding
This article is part of a special issue on women's health and empowerment, led and sponsored by the University of California Global Health Institute, Center of Expertise on Women's Health, Gender, and Empowerment. The broader systematic review on which this study is based was conducted under MEASURE Evaluation Phase III and the Health Policy Project funded by the United States Agency for International Development.

## Availability of data and materials
The authors do not wish to share the data at this time, given that four institutions are responsible for the data, and permission for making the data publicly available is needed from all institutions. However, data are available from the authors upon request and after permission is granted from the four institutions.

## About this supplement
This article has been published as part of BMC Pregnancy and Childbirth Volume 17 Supplement 2, 2017: Special issue on women's health, gender and empowerment. The full contents of the supplement are available online at https://bmcpregnancychildbirth.biomedcentral.com/articles/supplements/volume-17-supplement-2.

## Authors' contributions
MM and AM conceptualized the study and wrote the manuscript. MM, AM, and SP analyzed and interpreted the data. All authors read and approved the final manuscript.

## Ethics approval and consent to participate
Not applicable.

## Consent for publication
Not applicable.

## Competing interests
The authors declare that they have no competing interests.

## 
## Author details
[1]MEASURE Evaluation, Carolina Population Center, University of North Carolina at Chapel Hill, Chapel Hill, NC, USA. [2]WaterAid India, New Delhi, India. [3]Health Policy Plus (HP+), Palladium, Washington, DC, USA.

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
