## [Open peer review. (PDF 129 kb) · BMC Pregnancy and Childbirth]

Reviewer reports

Title: A review of measures of women's empowerment and related gender constructs in family planning and maternal health program evaluations in low and middle income countries

Reviewer 1: Mellissa Withers

I enjoyed reviewing this article and think it makes an important contribution to the field. I felt it was very well-written, scientifically sound, with relevance to the field. It builds more evidence for the need to give more attention and thought to the measurement of women's empowerment constructs. The methods were relatively complicated but the tables and figures helped to better understand the process. I would recommend this for publication.

- Major Compulsory Revisions

I think it would help to orient the reader more if some definitions were presented up front and some examples are given for these terms. For example, it would be helpful to explain terms such as "gender-integrated" in the very beginning of the paper. I believe the authors referred to this as programs that "actively seek to identify and integrate activities that address the role of gender dynamics." I feel that more information is needed for this. For readers who are not as familiar with this type of programs, it might help to give an example or two of such programs.

Also, in the results, I think it would be useful to refer to exact articles in the discussion of the different categories. For example, the authors say that nine articles that included any gender measurement examined a single gender domain. I believe the articles need to be cited here. Same for the articles that used scales (some used validated scales and others did not). I was curious about the qualitative articles. The authors state that the articles seldom specified the gender or women's-empowerment-specific dimensions or questions within the guides. Is this because they didn't include the guides at all? Or because they didn't even ask these questions directly? If the latter is the case, can the authors really count these articles as ones that measure gender-related constructs? Or did the gender-related findings simply emerge as part of the findings. I believe there is an important difference between the studies that intend to measure how gender might influence programs and those in which gender-related issues simply emerge later.

I also thought it might be helpful to discuss how many of the articles used secondary data such as the DHS versus primary data. Obviously, the use of secondary data will have major limitations in terms of what is available for gender-related measures.

I think it would also be helpful for the authors to describe some examples of the measurement of empowerment related to "sexual relations, reproduction, and childbirth."

I was also curious to know more about the programs. Were the majority microfinance based? Or cash vouchers? And what were the major outcomes that they were looking at (use of prenatal care, current use of FP, etc.)? This would have been very interesting too. I think this warrants some discussion in terms of looking at gender and its relationship with health outcomes. Certain health outcomes seem to lend themselves more to this type of programming. Or is it that these health outcomes are easier to measure as compared to outcomes such as social norms, community perceptions, etc.) The authors could have commented on this as well.

The discussion raised some very important points. I think the authors could have expanded this section to include some recommendations in terms of how to improve study design in the future in measuring the impact of gender on health outcomes. What are the major considerations in the conceptualization of these constructs? How can this be improved? While there was one short paragraph on this, I felt that this section could be developed more. Maybe a table or chart would also be useful.

I was also pleased to see a discussion about the potential unintended negative consequences of studies that aim to increase women's empowerment.

- Minor Essential Revisions

n/a

- Discretionary Revisions

I felt that the introduction started very abruptly without enough introduction to the topic. I would suggest just a few lines about the importance of the gender issues in MH and FP outcomes.

In the first paragraph, last line, the authors refer to empowerment and health outcomes but do not make the distinction of **women's** empowerment and women's health outcomes specifically, which I think should be clarified.

In the second paragraph, there is a discussion regarding women's empowerment in terms of

Level of interest

- An article of importance in its field

Quality of written English

- Acceptable

I declare that I have no competing interests.

Reviewer 2: Shari Dworkin

Dear Colleagues:

I have read your interesting paper that examines gender integrated programs and seeks to understand the extent to which gender is measured in the programs. The authors argue that without proper measurement of gender and empowerment constructs, the field will not be able to know if interventions focused on gender related content was the reason for program success (improved health outcomes; here family planning and maternal health). This is a paper that fits the goals of the special issue and has solid promise. I have many questions and suggestions for improvement in the paper. Overall, if these revisions can be tended to, I think the paper would make a strong contribution to the literature.

1. Qualitative studies that are included in this paper are not intended to measure constructs unless the studies explicitly state they are engaged in measurement--these studies are more often focused on meaning and participant experiences. This current paper is focused on measurement. Thus, I would suggest removing the qualitative studies from the "measures" studies if these studies explicitly did not intend to measure constructs. This is not to say that qualitative studies aren't important because they are--but they may not help answer the research question which is focused on whether gender integrated programs measuring gender related constructs.
2. The authors offer little critique of the existing measures at the end of the paper and this is a missed opportunity. For example, during the course of the paper, the authors note that there are many different levels of analysis for measures that can help the field move forward--and yet they note the measures that exist are largely concentrated at the individual and couple level. The authors should make comments about this at the end of the paper--about the need for broader conceptions of gender and women's empowerment and the need for measuring at different levels of analysis. Clearly, we need more measures at different levels instead of reducing most of women's empowerment to decision making at the individual level.
1. The authors do not define "gender integrated" programs in the abstract or early enough in the paper. In fact, it is not clear why the authors draw on the term gender integrated when it is not a term that is generally used in the scientific literature. Readers also do not know what sector the interventions are in that the authors are examining--are the interventions from NGOs, INGOs, CBOs, or academic-community partnerships? University trials? If the paper is truly ONLY about gender integrated programs from NGOs, ingos, etc and NOT the science base, the authors need to be very clear about that and explain the choice.
2. Early in the paper, the authors state that there are few reviews that examine gender measurement and interventions and the impact of intervening on gender inequality to impact on health-- they left out a few pieces that should be added: Dworkin, Treves Kagan & Lippman 2013 who examined the efficacy of gender transformative work with men to reduce violence and HIV risks. This a systematic review and the findings in the current paper often mention gender transformation and the gender equitable man scale, so this review should be added.

3. Continuing along the lines of point 3, the terms used in the science base to describe gender related interventions focused on health are the following: “gender sensitive” and “gender empowering” and “gender transformative.” The paper is therefore confusing because the authors use the term gender integrated. And, the authors do not use gender sensitive as a term or gender empowering as a term--but they do use gender transformative. I would say that readers of BMC will not know what gender integrated means because science based readers are accustomed to gender neutral, sensitive, empowering or transformative language. One more question here: Why leave out empowering but include transformative interventions in the review? Please define the terms you use.
4. There is no chart to show what the actual measures are (the level is noted but not the name of the scale) ---also while we know the level of analysis of the measure—we don't know the level of analysis of the intervention. In other words: Are the gender measures at the same level of analysis as the intervention? IF not, please show the level of analysis of the intervention too. If they match, say so? IF they don't match—comment in the conclusions about the mismatch of measures-- to press the measurement field further?
5. One large weakness was that the authors make no mention of the design of the studies. Not only is it important to point out that authors should measure gender or empowerment, but that the design of the study can affect what we can definitively say about the relationship between empowerment and or/gender and health. Are the interventions randomized trials that are examined in this paper? Are these Pre post test with no control group? Are these Quasi experimental designs that are pre post with a control group but are not a randomized design? Readers need to know this because this would affect our assessment of the strength of the evidence. There are no comments at all about the strength of the evidence in any of the interventions in which these measures are immersed. Of the 31 interventions—what % are randomized trials? If gender integrated programs aren't RCTs that would help the field more definitively assess the evidence, this may limit the creation of evidence on the links between women's empowerment and health and there could a call for more rigorous designs at the end of the paper.
6. The authors make comments about male involvement being potentially problematic— but the interventions mentioned in this part of the paper weren't gender transformative so it is not all that surprising that women's power would be negatively affected--perhaps the authors can distinguish between the effectiveness of male involvement and gender transformative programs because they are quite different in terms of the impact on gendered power relations.
7. What proportion of the total measures were using the gender equitable man scale? IF most studies measured women's empowerment through measuring whether programs made men more equitable, what does this say about how well women's empowerment work? This seems significant to comment on.
8. Also—think—are the gender related measures that are covered in this paper indicators of empowerment or something else? Help readers to nuance this thinking. The authors conflate the word gender and the word empowerment throughout the paper—there are gender related constructs that are unrelated to empowerment and there are gender

related constructs that are related to empowerment. Which is being focused on in this paper? To what extent are gender integrated programs focused on empowerment? . Why assume that gender integration means gendered empowerment? To what extent does it?

9. The authors didn't provide enough clarity for readers on whether women are the focus of the interventions examined or men or both
10. One thing that was totally left out: are there fabulous measures that interventions should be including but are not including? Maybe the measures are out there and exist more so but programs aren't including them in interventions. Can you mention some of the great measures that are out there? For example—the sexual relationship power scale exists and is validated and has been modified in many settings--- is this not in the programs examined related to HIV? What I am getting at is that in the conclusions, perhaps 1 point should be “include the existing measures” in the programs and try to intervene on these processes? Authors do mention the reproductive autonomy scale—it is very new—so perhaps researchers and practitioners should be called upon at the end of the paper to use these measures in interventions and modify them for the setting.
11. The authors seem to call for standardization of measures but it seems to be the case that the authors found very few measures in the programs. Maybe they should be calling for MORE measure development and modification at this point and not standardization?
12. Can the authors list the LMIC countries somewhere or provide a link for readers so all readers know which countries are included if they don't recall all countries?
13. The discussion of GBV on page 8 needed reconsideration—this is a health outcome so I'm not sure why the authors discuss it as an empowerment domain. However—I think it would be highly relevant to include it as a health outcome of interest because this IS a maternal health outcome. And, there are existing gender sensitive and empowering interventions that measure gender related empowerment and do work to reduce violence against women which is a reproductive health and maternal health issue. See Julia Kim's and Paul Pronyk's analysis of the IMAGE study. See comment 18.
14. The authors can strengthen the conclusions significantly if taking up the points in this review more. One more: They do not recognize that empowerment actually can mean different things in different contexts and has different relationships to health outcomes and thus may require unique or modified measures in different places. Measure development may be needed that is sensitive to the different processes and outcomes related to empowerment. AS noted, the authors call for standardization...without more measure development this doesn't seem like a reasonable place to end the paper
15. The discussion of self indulgent freedom and productive value seemed to logically stray from the focus of the paper and also conceptually extraneous to introduce at this stage of the article. The authors introduce Kabeer's more accepted definition on the front end of the paper that is often used in global health studies but then introduce this less commonly used concepts at the end focused on self indulgent freedom. I would suggest an attempt to be more consistent and return to the common definition by Kabeer raised at the outset in the paper....return to it—are measures actualizing our accepted definitions? If not—do we need more measures? In fact—Kabeer was focused on how

resources shape agency and empowerment. Do the interventions covered in the paper focus on this? If not, maybe Kabeer's definition needs measure development and content development in interventions?

16. Women are not a homogeneous group. The authors make no mention of the samples of women in these studies or the sample size. Are they of a particular race? Religion? Age? Sexual Orientation? Is measure development focused on some of these populations of women and not other populations of women? IF so: what other measures are needed to truly understand more about all women in terms of empowerment and health?
17. The authors state that they are interested in measuring women's empowerment and examining gender integrated programs but they make little mention of women's rights or measuring rights. There are new articles out that discuss how to measure rights and rights based processes—these can be used in interventions and programs too. Did any interventions/programs examined in this review focus on rights or measure rights? If not why do you think this is? Shall the authors call for more work on measuring rights and implementing gender integrated interventions that on rights based processes to improve health? Especially see: Polet's work in *Health and Human Rights* in 2015. Also see Gruskin and Ferguson's work.
18. There were not that many peer reviewed articles on these topics. There were more articles in the grey literature. Can the authors comment on this finding? Im not sure the authors have covered all science based interventions either. They say HIV and STDs can be an outcome for the inclusion criteria—but the randomized controlled trial by Julia Kim and Paul Pronyk isn't mentioned which measured women's empowerment in South Africa—see this work. Also see the interventions mentioned and listed in the Dworkin Treves Kagan and Lippman review too.
19. Was one area of the literature stronger than the other in some way in terms of measures? Were maternal health programs not as far along while family planning was-- in terms of measurement—or the reverse? In what ways? Can you differentiate at all for readers--instead of lumping these interventions together?

SMALL COMMENTS

1. The abstract was not very clear—define gender integrated—define the evidence/the study designs more.
2. Cut the word ubiquitous from the paper—I think the authors mean “consistent”
3. Its not clear why authors refer to the developing country context in line 214 page 8 but cite a US study in the cites to explain this point

Level of interest

- An article of importance in its field

Quality of written English

- Needs some language corrections before being published

Declaration of competing interests

NO COMPETING INTERESTS

Reviewer reports – 2nd round

Reviewer: Mellissa Withers

After reviewing the extensive comments from the three reviewers, and the authors responses, I feel like they have done a great job of revising this paper. They had a lot of comments to address but I think the paper was significantly strengthened. (I was also pleased to see that Shari and I had many similar comments on the paper.)

I also read the new version last night and feel like it is ready to be accepted.

Let me know if you have any questions.

Reviewer: Shari Dworkin

The paper is responsive to reviewer comments and is substantially improved. I have minor comments only.

1. It is not clear in the introduction, when the authors are describing gender accommodating interventions--what "work around" means or what "adjust" means. This needs clarification.
2. On page 5, the authors use the word "health outcomes" twice, but they note in their cover letter that they will state "family planning and maternal health outcomes"--I would ask them to specify the health outcomes as such on page 5 in these instances.
3. It is not clear what "health and agency program" means on page 6
4. Change "a third" to "one third" throughout
5. There are 2 routes to empowering women. Programs that work with women are one way. Another way is gender transformative work with men to move men in the direction of more gender equality. The latter slips away from the paper and needs to be discussed in the discussion section of the paper. If other health fields such as HIV and violence work are further along than FP and MH in this regard, then what might we apply from that work to MH/FP outcomes? Wouldn't this be an important area for further research?
6. Minor edits are needed throughout; I leave this to the editors.

Response to reviewers – 2nd round

Chapel Hill, NC, January 25, 2017

Dear editorial board of BMC Pregnancy and Childbirth Special Issue on Women's Health and Empowerment:

Please find attached our third draft of the manuscript: "A review of measures of women's empowerment and related gender constructs in family planning and maternal health program evaluations in low and middle income countries." We thank the reviewers for their very fast turn-around on the second draft of our manuscript, and for their additional comments. We have addressed all comments within the revised manuscript and provided detailed responses below. We look forward to hearing from the editors.

Sincerely,

Mahua Mandal

Responses Addressing Comments from Managing Editor: Ushma Upadhyay

1. I agree with Shari Dworkins concern with the use of “health outcomes.” It is too broad a category and needs clarification in a few places.

Response (R): We have addressed this comment throughout the paper, including on pages 2, 5, and 12.

2. Page 10, I suggesting changing the word last to final because the meaning is more commonly understood. So it says: (e.g. if the decisions are made mostly by the man, woman, or jointly; or who has the final say)

R: We have changed this wording.

3. Please review your references carefully and check for errors. Glancing quickly, I see errors in the following references: 1, 3, and 4

R: We have corrected the references.

4. Figure 2. There is a typo in Latin America

R: We have corrected this.

5. I suggest adding the citation numbers to Table 1.

R: We have added the citations to Table 1.

6. Table 1: In the intervention column, some of them are not immediately clear what type of intervention it was. Can you be more specific about the following ones: “Economic well-being of families” “Change social norms” and “Women’s empowerment”.

R: We’ve addressed this in Table 1 and provided more information on the interventions

7. Table 1: The results column is not quite clear what the effect sizes represent. Is it being in the intervention/program vs. a control group? If so, please add this in a footnote.

R: We have clarified what the effect sizes represent through the addition of footnotes.

Responses Addressing Comments by Reviewer: Shari Dworkin

The paper is responsive to reviewer comments and is substantially improved. I have minor comments only.

1. It is not clear in the introduction, when the authors are describing gender accommodating interventions--what "work around" means or what "adjust" means. This needs clarification.

R: We have added an example to the definitions to improve clarity. On page 3, we state: "Gender transformative approaches actively strive to challenge and change inequalities with the aim of achieving gender equality while promoting health. These approaches encourage critical awareness of gender roles and norms; challenge the distribution of resources and allocation of responsibilities between men and women; address power relationships between men and women; and promote the position of women. For example, a national policy may require husbands to accompany their wives to a FP clinic in order for women to get contraception. A gender transformative intervention would work to change this policy in order for women to access contraception without their husbands. Gender accommodating interventions work around inequitable gender norms, roles, and relationships or adjust for these inequalities. While these approaches do not actively seek to change norms and inequalities, they strive to limit the harmful impact of interventions on gender relations and harmful impact of gender norms and inequalities on health outcomes (1, 2). Using the same example above, a gender accommodating intervention would increase knowledge of the existence of this policy among couples and encourage husbands to accompany their wives to clinics so women can access contraception."

2. On page 5, the authors use the word "health outcomes" twice, but they note in their cover letter that they will state "family planning and maternal health outcomes"--I would ask them to specify the health outcomes as such on page 5 in these instances.

R: We have changed "health outcomes" to "FP and MH outcomes" on page 5.

3. It is not clear what "health and agency program" means on page 6

R: We have rephrased this to state: "Only one article described its intervention as primarily focused on increased women's autonomy (21)."

4. Change "a third" to "one third" throughout

R: We have changed this wording

5. There are 2 routes to empowering women. Programs that work with women are one way. Another way is gender transformative work with men to move men in the direction of more gender equality. The latter slips away from the paper and needs to be discussed in the discussion section of the paper. If other health fields such as HIV and violence work are further along than FP and MH in this regard, then what might we apply from that work to MH/FP outcomes? Wouldn't this be an important area for further research?

R: We focused on the measurement piece and added the following to the discussion: “Fifth, the FP and MH evaluation field should look to studies of male-focused gender transformative HIV and violence interventions to identify and adapt measures of male engagement and related gender constructs. Measures of normative change among males, including attitudes around gender roles and masculinity, may be particularly important in evaluations of FP and MH interventions that involve men with the explicit purpose of increasing gender equity and women’s empowerment.”

6. Minor edits are needed throughout; I leave this to the editors.

R: We have edited the manuscript